# Dual-comb spectroscopy of laser-induced plasmas

Jenna Bergevin[1], Tsung-Han Wu[1], Jeremy Yeak[2], Brian E. Brumfield[3], Sivanandan S. Harilal[3], Mark C. Phillips[3] & R. Jason Jones[1]

Dual-comb spectroscopy has become a powerful spectroscopic technique in applications that rely on its broad spectral coverage combined with high frequency resolution capabilities. Experiments to date have primarily focused on detection and analysis of multiple gas species under semi-static conditions, with applications ranging from environmental monitoring of greenhouse gases to high-resolution molecular spectroscopy. Here, we utilize dual-comb spectroscopy to demonstrate broadband, high-resolution, and time-resolved measurements in a laser-induced plasma. As a demonstration, we simultaneously detect trace amounts of Rb and K in solid samples with a single laser ablation shot, with transitions separated by over 6 THz (13 nm) and spectral resolution sufficient to resolve isotopic and ground state hyperfine splittings of the Rb $D_2$ line. This new spectroscopic approach offers the broad spectral coverage found in the powerful techniques of laser-induced breakdown spectroscopy (LIBS) while providing the high-resolution and accuracy of cw laser-based spectroscopies.

[1] College of Optical Sciences, University of Arizona, 1630 E University Blvd, Tucson, AZ 85721, USA. [2] PM&AM Research, Tucson, AZ 85719, USA. [3] Pacific Northwest National Laboratory, Richland, WA 99352, USA. Correspondence and requests for materials should be addressed to R.J.J. (email: rjjones@optics.arizona.edu)

Laser-induced plasmas provide a versatile and non-contact means to apply the powerful tools of optical spectroscopy in the analysis of solid materials. When the intensity of an incident laser pulse is sufficiently high, a plasma is generated, ablating a small amount of material above the sample surface. The resulting ions, atoms, and molecules within the evolving plasma plume can be detected optically using either emission or absorption techniques. The most common optical spectroscopy technique for laser ablation plumes is optical emission spectroscopy, typically called laser-induced breakdown spectroscopy (LIBS)[1,2]. In this technique, the emission spectrum from electronically excited ions, atoms, and molecules is recorded. LIBS has had a significant impact in a broad range of fields. For example many geological applications exist for studying the composition of rocks, minerals, and soils[3], including the planetary exploration of Mars by the NASA rover Curiosity[4]. LIBS has also been applied for the characterization of nuclear materials and their isotopic concentrations[5], needed for monitoring nuclear waste and nuclear fuel production for its use in both civilian and military applications. Other applications are found in areas ranging from pharmaceutical quality control[6], monitoring of industrial processes[7], forensic science[8], and for trace detection of nanoparticles used in medical applications[9], to name a few.

In this work, we apply the technique of dual frequency comb spectroscopy to optically probe laser-induced plasmas. Dual-comb spectroscopy (DCS) has become a valuable tool for broadband and high-resolution spectroscopic measurements. We demonstrate the potential impact of this approach in the analysis of solid materials by simultaneously detecting trace amounts of Rb and K in solid samples using a single laser ablation shot while still providing sufficient spectral resolution to observe the isotopic and ground state hyperfine splittings of the Rb $D_2$ line. The broadband, sensitive, and high spectral and temporal resolution of this approach can enable the capability to optically identify and track multiple atomic, ionic, and molecular species present in the dynamic plasma environment.

## Results

**Laser ablation spectroscopy.** To obtain the maximum amount of information about the composition of a solid sample via optical probing of a laser ablation plume, it is desirable to utilize a large spectral bandwidth so that multiple ionic, atomic, molecular, and continuum transitions can be identified. At the same time, a high spectral resolution is needed to distinguish closely spaced transitions, to resolve small isotopic shifts, to measure intrinsic linewidths of transitions, and to minimize spectral interferences from multiple species. Other desirable aspects of an optical spectroscopic measurement are rapid detection, time and space resolution, and simultaneous acquisition of multiple wavelengths. Unfortunately, it is difficult to satisfy all requirements simultaneously and thus compromises must be made based on available spectroscopic instrumentation.

In LIBS, a dispersive spectrograph combined with an intensified-CCD (ICCD) camera is often used to record the optical emission. It is straightforward to measure time-gated emission spectra from laser ablation plasmas with the ICCD. However, this technique requires a trade-off in spectral bandwidth versus spectral resolution based on diffraction grating selection and ICCD array size. Thus, a low- to moderate-resolution spectrum can be obtained over a large spectral bandwidth, or a high-resolution spectrum can be obtained over a reduced spectral bandwidth. For example, a typical LIBS spectrum measured using a 0.5 m Czerny-Turner spectrograph with 2400 g/mm grating could provide a spectral resolution of ≈20 pm with ≈5 nm spectral bandwidth. If the emission intensity is high enough, spectra may be obtained from single ablation shots without additional averaging.

A fundamental limitation of optical emission spectroscopy is the requirement for electronic excitation to generate a measurable emission signal. The relative populations of ionized atoms, neutral atoms, and molecules vary as the plasma evolves, and are highly dependent on the ablation conditions. Nevertheless, for any given transition the emission intensity increases with excitation temperature that is higher at earlier times in the plasma evolution. Unfortunately, higher temperature and electron density at early times in plasma evolution also leads to increased spectral linewidths, primarily via Stark broadening[2]. Linewidths are decreased at later times in the plasma evolution, but the emission intensity also decreases at these times. Thus, for optical emission spectroscopy techniques, maximizing signal intensity and minimizing spectral linewidths are conflicting requirements.

Absorption spectroscopy techniques applied to laser ablation plumes measure lower-state populations and thus do not rely on electronic excitation of species. As a result, absorption may probe later times in the plasma evolution during which temperatures are cooler and spectral linewidths are correspondingly narrower. For the conditions used here (ns ablation pulses, low background pressures), the plasma emission lifetime typically lasts a few microseconds. However, the presence of ground state atomic neutrals can last several hundreds of microseconds.

Absorption spectroscopy using broadband incoherent sources and a dispersive spectrograph has been used for probing laser ablation plumes, but suffers the same limitations on spectral bandwidth and resolution as optical emission spectroscopy[10]. Tunable cw lasers provide a means to probe laser ablation plumes via laser absorption spectroscopy (LAS)[11-16] or laser-induced fluorescence (LIF)[15-20]. The spectral linewidths of cw lasers (<10 MHz) are much smaller than the intrinsic linewidths of most transitions. As a result, spectra may be measured in the laser ablation plume by LAS or LIF without effects of instrumental broadening. For example, atomic absorption spectral linewidths measured in laser ablation plumes can be <2 GHz (2 pm)[14,19,20]. However, when performing LAS or LIF using cw lasers, the laser wavelength must be stepped or scanned across an absorption feature to record a spectrum. The scanning requirement places a limit on the spectral bandwidth that may be acquired in a given time period for LAS or LIF. Moreover, acquisition typically occurs over multiple ablation pulses while the laser is scanned, making single-shot measurements impractical. Performing multiplexed wavelength measurements on a single ablation shot would provide great benefits in noise reduction, consistent plasma conditions over all wavelengths, and increased spectral coverage.

**Dual-comb spectroscopy.** Femtosecond (fs) frequency combs offer a new and powerful tool for probing laser ablation plasmas. Frequency combs have revolutionized the field of precision frequency spectroscopy and enabled the development of next-generation atomic clocks based on optical transitions[21]. Frequency combs are based on the ability to generate trains of optical pulses with a well-defined phase between subsequent pulses. This pulse-to-pulse phase coherence results in an optical spectrum consisting of narrow comb teeth separated precisely by the pulse repetition rate (e.g. 100 MHz). Furthermore, the absolute optical frequency of each comb tooth can be determined with high accuracy. Given the short pulse durations that can be obtained directly from many laser systems (<10 fs), the bandwidth of these optical frequency combs can easily extend beyond 100 nm. In addition, thanks to the high peak powers provided by the pulses, nonlinear frequency conversion techniques can be utilized to extend the frequency comb coverage to over an optical octave[22].

This makes the achievable spectral coverage provided by the frequency comb comparable to that found in many LIBS systems.

Spectroscopic techniques often use a tunable cw laser to probe an atomic or molecular transition, with the frequency comb used to subsequently measure the absolute frequency of the cw laser. In direct frequency comb spectroscopy, the cw laser is not needed, and the transitions are probed directly with the broad spectrum of the frequency comb. A variety of configurations can be used for direct frequency comb spectroscopy[23]. In particular, DCS[24] combines the strengths of both spectrometer-based broadband spectroscopy and tunable laser spectroscopy to simultaneously provide high frequency resolution and accuracy, high sensitivity, and extremely broad spectral coverage with the use of a single photodiode as detector. Operation of DCS is similar to Fourier-transform spectroscopy. In Fourier-transform spectroscopy one arm of an interferometer is scanned in time to obtain an interferogram, which is then Fourier-transformed to obtain an absorption (transmission) spectrum in the frequency domain. In DCS[24], two fs frequency combs are used, with a slight difference in repetition rate of $\Delta f_{rep}$. One of the fs combs is used as the spectroscopic probe beam and the second as a local oscillator. When the pulse trains are overlapped (before or after the sample) and interfere on a photodiode, the time delay between pairs of pulses from each laser will change by discrete steps with each successive pulse pair, resulting in a discretely sampled interferogram. As in Fourier-transform spectroscopy, the Fourier transform of this interferogram yields the optical absorption spectrum of the sample. This replaces the slow mechanical delay arm used in Fourier-transform spectroscopy with an optical delay that can be scanned rapidly and with unprecedented accuracy.

Previous work utilizing DCS has focused primarily on spectroscopy of systems under semi-static conditions, including atmospheric sensing of greenhouse gases[25] and high-resolution spectroscopy of gas cell-based samples[26–28]. Time-resolved direct comb spectroscopy with a single frequency comb has been performed using highly dispersive spectrometers to resolve individual comb teeth (e.g. detection of transient free radicals in chemical reactions[29]), but more recent work has begun to exploit the rapid scanning capability of dual-comb systems for time-resolved measurements[30,31]. Time-resolved DCS applied to transient events such as laser plasmas combines the broadband capability of LIBS with the high spectral resolution of tunable LAS. Multiple atomic or molecular absorption lines can then be measured simultaneously with extremely high spectral resolution using broadband frequency combs, promising greatly improved plasma characterization and higher detection confidence for species in the plasma.

In this paper, we present results using dual frequency comb spectroscopy to study laser-induced plasmas. As shown in Fig. 1, the pulse trains from two fs combs were combined on a beam splitter, with one pair of overlapped beams focused into a vacuum chamber using a 20 cm focal length lens just above the surface of either an NIST glass sample (SRM610) or a mica sample (lepidolite). The NIST SRM610 glass contained 425 ppm of Rb and 461 ppm of K, while the mica sample had not been analyzed for chemical content, but contained approximately 3% or less of Rb. A Q-switched Nd:YAG laser (Continuum Surelite II) operating near 10 Hz with 85 mJ in a 10 ns pulse was focused onto the sample to generate a plasma in the vacuum chamber. The Nd:YAG was triggered at 10 Hz by dividing $\Delta f_{rep}$ by 50. We recorded the full interferogram between the probe and local oscillator pulse trains after each laser ablation pulse with a delay of the interferogram set at approximately 220 μs. This time delay was chosen to minimize the effects of Stark and Doppler broadening at earlier times in the plasma evolution. The second pair of beams from the beam splitter were sent through an Rb reference cell. The interferograms from both beams were detected on identical silicon photodiodes. Simultaneously recording interferograms from the ablation chamber and the Rb cell enabled comparison between line broadening and shot-to-shot variations from the laser-induced plasma and the Doppler-broadened and static room temperature Rb cell.

Figure 2 shows the broadband absorption spectrum recorded from a single laser ablation shot of the NIST glass. The data was acquired in a 440 μs time window with $\Delta f_{rep} = 520$ Hz, corresponding to a measurement-limited resolution of 0.53 GHz (1 pm). The absorption spectrum clearly shows the presence of both Rb (780.0 nm) and K (769.90 and 766.49 nm) atomic absorption lines. The inset shows a zoomed-in portion of the $5S_{1/2}-5P_{3/2}$ Rb $D_2$ line, with the four absorption peaks from the Rb $D_2$ ground state hyperfine structure and isotopic shift of $^{85}$Rb versus $^{87}$Rb identified. Our approach enables us to identify multiple species in a single laser ablation shot covering optical bandwidths (10 nm) not feasible with a cw laser source, while still providing GHz-level spectral resolution.

In Fig. 3 we show a portion of the absorption spectrum recorded simultaneously in (a) the laser-induced plasma from the mica sample and (b) the Rb cell, each averaged over three laser ablation shots. By averaging over only three laser ablation shots, we demonstrate the improved optical resolution and signal to noise levels that can be achieved. The four absorption peaks from the Rb $D_2$ are now more clearly resolved, with linewidths measured in the plasma only slightly greater than that from the Rb cell measurement. That the Rb spectrum from the plasma could be measured by the frequency comb nearly as cleanly as that of the cell-based measurement demonstrates the promise of DCS for probing laser-induced plasmas with high-resolution and accuracy for the detection of atomic, molecular, and ionic species, and their isotopic splittings, across large spectral bandwidths.

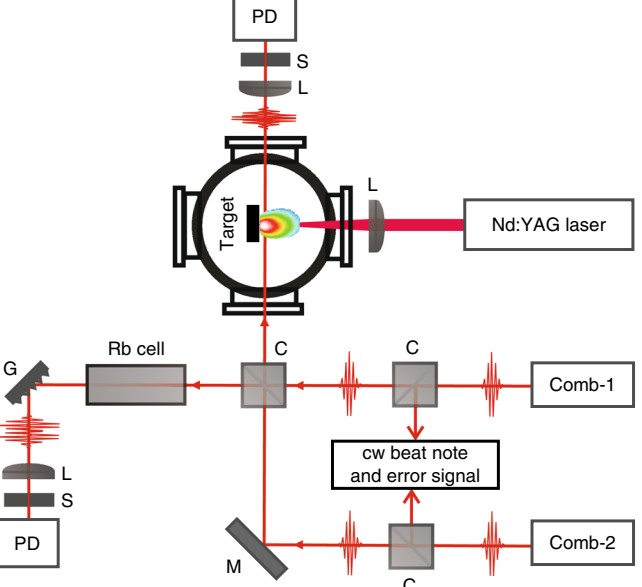

**Fig. 1** Schematic of experimental layout. A pair of modelocked fs lasers, Comb-1 and Comb-2, are actively stabilized to a common cw laser using optical phase-locked loops to enable high-resolution dual-comb spectroscopy of a laser-induced plasma. The laser beams are spatially overlapped on a beamsplitter and simultaneously probe a room temperature Rb cell that serves as a reference spectrum, and a solid ablation target containing trace amounts of Rb and K inside a vacuum chamber. Dashed lines indicate electrical phase-lock feedback loop. *PD* photodiode, *G* grating, *S* slit, *M* mirror, *L* lens, *C* cube beam splitter

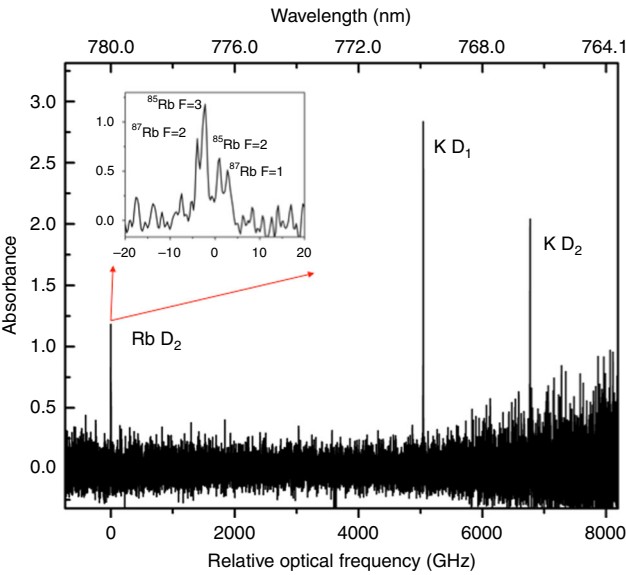

**Fig. 2** Broadband single-shot absorption measurement of the laser-induced plasma. The absorption spectrum of the Rb $D_2$ line and K $D_1$ and $D_2$ lines, separated by over 6 THz, are simultaneously observed from a calibrated sample of SRM610 NIST glass following the laser ablation pulse. The inset shows a zoomed-in view of the $5S_{1/2}-5P_{3/2}$ Rb $D_2$ line, demonstrating the ability of DCS to simultaneously provide high spectral resolution by resolving the four absorption peaks from the Rb $D_2$ ground state hyperfine structure and isotopic shift of $^{85}$Rb versus $^{87}$Rb. Optical resolution: 0.53 GHz (1 pm), $\Delta f_{rep}$: 520 Hz, pressure: 51 torr. Labels indicate Rb isotope and ground state hyperfine level of the transition

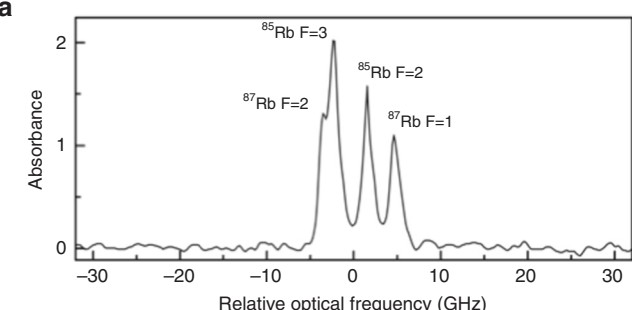

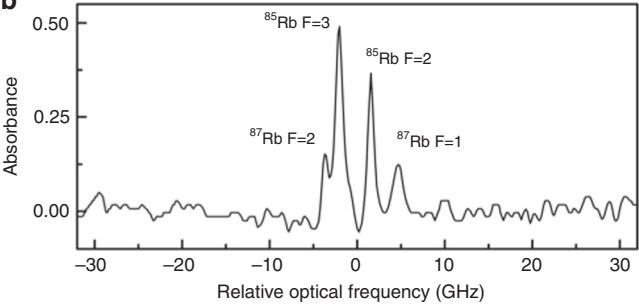

**Fig. 3** Comparison of Rb cell versus laser plasma absorption spectrum. Simultaneous measurement of the Rb $D_2$ lines from the **a** laser-induced plasma and **b** Rb cell. Each spectrum is the average of three laser ablation shots. The calculated measurement resolution was 0.18 GHz (0.4 pm). $\Delta f_{rep}$: 140 Hz, pressure: 1 torr

## Discussion

The broadband and high-resolution spectrum of multiple atomic species measured in a single laser ablation shot as demonstrated

here using DCS would be extremely difficult to achieve using other techniques. For example, in LIBS, obtaining sufficient spectral resolution to resolve intrinsic spectral linewidths in a laser ablation plume presents significant challenges and requires very large spectrographs, further reducing spectral bandwidth. Intrinsic LIBS emission linewidths of atomic transitions are often <10 pm, especially for heavy elements such as U[32,33]. Approaches have been used to increase spectral resolution in LIBS measurements, such as combining a Fabry−Perot etalon with a spectrograph[32,34]. Echelle spectrographs[33] are capable of measuring broad spectral bandwidths at high spectral resolution. For example, an echelle spectrograph using a CCD can measure over a spectral bandwidth of 200–1000 nm at a resolving power of 44,000, corresponding to a spectral resolution of 5–20 pm[33], which would still be insufficient to measure the Doppler-limited linewidths and hyperfine splittings demonstrated here. In addition, the low optical throughput of such instruments can make some LIBS measurements challenging, especially when time-resolved or single-shot measurements are desired.

In LAS or LIF measurements that use cw lasers to achieve high spectral resolution, the limited scanning range presents a serious constraint on optical bandwidth. For example, an external cavity diode laser operating at 780 nm may provide 50 GHz of mode-hop-free tuning range which is usually only enough to measure the absorption spectrum of a single transition at a time. LAS and LIF also face challenges when applied to laser plasma measurements due to the highly transient and spatially varying plasma environment. The characteristics and shot-to-shot repeatability of the laser plasma (e.g. density, temperature, species evolution) depend critically on multiple factors, including sample surface conditions and possible multi-shot modifications, pulse energy, pulse duration, background pressure and composition to name a few[2,35]. Variations in ablation properties during acquisition with a cw laser over multiple shots can lead to noise in the absorption spectrum or uncertainties in spectral parameters due to changing plasma physical conditions, especially at higher ambient pressures. Techniques have been developed to reduce the effects of this shot-to-shot ablation noise using differential absorption[14] or signal normalization by emission[20]. However, in the DCS measurements demonstrated here, all frequencies of the fs comb interact with the plasma simultaneously. The noise seen when scanning a tunable laser over multiple ablation shots can in principle be minimized with DCS. Even if ablation properties change over multiple ablation shots, this noise will appear as spectrally flat in DCS, which can be averaged more effectively than in tunable LAS.

The signal-to-noise (s/n) of the absorption spectrum demonstrated here can be further improved using a variety of standard techniques already implemented in other DCS experiments. For example, a reference beam can be used to differentiate background noise from the measured absorption signal[36]. In the current experiments, a time-interleaved reference beam is inherently available from the probe pulses arriving in between laser ablation shots when there is no plasma present. Modifications in the current timing electronics and data acquisition processing can use the information from these temporal slices of the pulse train to subtract out residual background noise and improve the detection sensitivity.

The initial experiments reported here were performed with relatively long integration times considering the highly transient nature of laser-produced plasmas. However, the changes in plasma properties are more drastic at early times of its evolution, and much less during the delayed temporal window of the present measurements. This enabled the measurement of relatively narrow lineshapes not severely limited by the Stark or excessive Doppler broadening that occurs at shorter timescales. In the

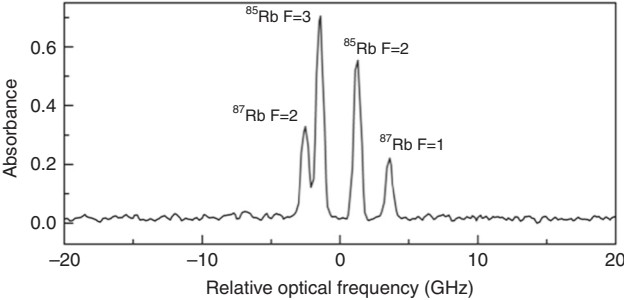

**Fig. 4** Doppler-limited absorption spectrum of Rb $D_2$ lines. The spectrum from the gas cell shows the ground state hyperfine splitting of both $^{85}$Rb and $^{87}$Rb. The spectrum is the average of 200 measurements with a calculated measurement resolution of 0.13 GHz

current experiment, we are also able to simultaneously detect the Rb and K transitions with a time window down to only ≈10 μs. In this case, the broadband spectrum (not shown) looks nearly identical to that as shown in Fig. 2. However, the hyperfine and isotopic shift of the Rb $D_2$ line is not resolved due to the reduced measurement-limited spectral resolution. Increasing $\Delta f_{rep}$ can improve the time-resolution without sacrificing optical resolution. However, this comes at the cost of detection sensitivity and measurement spectral bandwidth[37]. Probing the plasma with increased time-resolution utilizing higher repetition rate laser sources can be helpful in future work to study the dynamic evolution of the ionic, atomic, and molecular species within the plasma.

In conclusion, we have demonstrated DCS of laser-induced plasmas. We simultaneously measure trace amounts of Rb and K in a single laser ablation shot while resolving the hyperfine and isotopic shift of the Rb $D_2$ line with sub-GHz level resolution. The spectral resolution, broad spectral coverage, and rapid measurement capability of DCS can enable identification and tracking of multiple ionic, atomic, and molecular species in the evolving plasma. The spectral coverage of DCS can be easily extended to almost any portion of the electromagnetic spectrum. For example, with use of nonlinear optical fibers for supercontinuum generation, DCS laser systems can measure optical spectra spanning 100's of nanometers[22]. Further spectral coverage can be achieved with sum[28] and difference frequency generation[38], and even intra-cavity high harmonic generation[39,40] to the vacuum and extreme-ultraviolet. Our approach to time-resolved DCS can also be applied to study other dynamic systems such as kinetics in chemical reactions[29] and pulsed-detonation combustion[41]. The results demonstrated here lay the ground work for future studies exploiting the properties of the fs frequency comb to probe laser-induced plasmas, with the potential to impact a wide range of applications in the optical analysis of solid materials.

## Methods

**Experimental set-up.** The dual-comb system used here was provided by PM&AM Research (model PMAM-780-TR-DCS-0.120 GHz) and was developed in a collaborative effort to investigate isotopic ratios of radiological materials utilizing laser-induced plasmas. It is composed of two Kerr lens mode-locked Ti:sapphire lasers each pumped at 4.5 W from a common 532 nm laser source to minimize pump-induced noise between the oscillators. The spectrum of both lasers was centered near 780 nm with bandwidths spanning 760−800 nm, adjustable with the use of intracavity prisms. The approximately 120 MHz repetition rate of each laser could be adjusted using translation stages and piezo-electric transducers (PZTs) attached to each cavity end mirror. The repetition rate difference between the lasers, $\Delta f_{rep}$, was typically set between 100 and 500 Hz. To establish phase coherence between the comb modes near 780 nm, a free-running but slow drifting monolithic DBR laser at 780 nm was used (Vescent D2-100) as a transfer oscillator. The cw laser was locked directly to Comb 1 using fast servo electronics fed back to the cw laser current to lock the beatnote between the two lasers, $f_{b1}$, to a stable RF reference. The beatnote between a mode of Comb 2 and the cw laser, $f_{b2}$, was also detected.

Due to the linewidth of the cw laser, direct stabilization of $f_{b2}$ to an RF reference was insufficient to establish phase coherence between modes of Comb 1 and Comb 2 near 780 nm. To eliminate the contribution of noise from the transfer oscillator, Comb 2 was stabilized to the difference between the beatnotes, $f_{b1} - f_{b2}$[42] using a second fast PZT in Comb 2. This enabled an effectively tight phase lock between Comb 1 and Comb 2 for the modes near 780 nm that was independent of noise from the cw laser, and sufficient for averaging over multiple interferograms for time scales on the order of 1 s. For the Doppler-limited linewidths measured here, no additional locking of either comb offset frequency was needed.

**Rb cell reference.** To demonstrate the performance of the DCS system and to serve as a real-time reference for the plasma-based experiment, we first measured the Rb $D_2$ transitions near 780 nm in a 7-cm long glass cell containing a natural abundance of Rb atoms (72% $^{85}$Rb and 28% $^{87}$Rb). The pulse trains from each fs comb were combined before the Rb cell on a beam splitter and detected with a silicon photodiode after diffracting off a grating (see Fig. 1). The grating was used to improve the retrieved signal-to-noise levels by eliminating shot-noise contributions from unused portions of the laser spectrum[37]. The resulting interferograms were recorded in the time domain with a 12-bit data acquisition card at 125 Ms/s. The Rb absorption spectrum shown in Fig. 4 is the result of 200 recorded interferograms that were individually Fourier transformed and then averaged in the frequency domain. For this data set, with $\Delta f_{rep} = 500$ Hz, the 1.8 ms acquisition time per interferogram provided a 130 MHz (0.3 pm) optical resolution. Both Rb isotopes and their ground state hyperfine splittings are easily resolved, limited by Doppler broadening from the room temperature gas cell.

**Data availability.** All relevant data that supports our experimental findings is available from the corresponding author upon reasonable request.

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

## Acknowledgements

This material is based upon work supported by the Air Force Office of Scientific Research under award number FA9550-15-1-0091, the National Nuclear Security Administration, Defense Nuclear Nonproliferation R&D Office, and the Department of Energy under Award Number DE-SC0004311 and Physics, Materials and Applied Mathematics Research L.L.C. The Pacific Northwest National Laboratory is operated for the U.S. Department of Energy (DOE) by the Battelle Memorial Institute under Contract No. DE-AC05-76RL01830. J.B. would like to thank the UA/NASA Space Grant Program for funding during the course of this research.

## Author contributions

J.B. and T.-H.W. contributed equally to this work. R.J.J., T.-H.W., J.Y., M.C.P., and S.S.H. conceived the experiments. Experiments were performed by R.J.J., J.B.,T.-H.W., J.Y., and M.C.P. R.J.J., J.B., T.-H.W., J.Y., M.C.P., B.E.B., and S.S.H. contributed to the analysis of the data and interpretation of the results. R.J.J., J.B., M.C.P., and S.S.H. wrote the manuscript with comments from all authors.

## Additional information

**Competing interests:** The authors declare no competing interests.

