## [Peer Review File(PDF 195 kb) · Nature Communications]

Reviewers' comments:

Reviewer #1 (Remarks to the Author):

This is interesting and well written paper. It deal with the use of dual dual-comb spectroscopy to demonstrate broadband, high-resolution, and time-resolved measurements in a laser induced plasmas.

I have two main concern about this paper:

1. A short proceedings publications by the same authors, Jenna Bergevin et al, "Dual-Comb Spectroscopy of Laser-Induced Plasmas" in CLEO 2017 (SW1L.2), contains very similar context as the current paper, including the same title. I am concerned about the novelty of the work, and if copyright issues may arise.
2. This is a communication and the paper will not be long, but the author spend very little time addressing the underlying physics and chemistry and instead focuses the previous literature and technology potential. Importantly the data interpretation seems rushed. This paper would benefit from a more extended discussions of the Rb D2 absorption spectra.

Reviewer #2 (Remarks to the Author):

This is a paper based on high-quality experimental work, of considerable novelty. Therefore, in terms scientific content I have no specific comments. However, the presentation is somewhat lacking and needs some improvement.

Page 1, middle left column: In my opinion, the word "thermally-excited" is incorrect terminology. It implies that the excitation mechanisms in the LIBS plasma are primarily due to atomic collisions in a hot gas, which is far from the truth. There are numerous other excitation mechanisms, e.g. penning ionisation and asymmetric charge transfer. I therefore suggest using "electronically excited" throughout the text. This refers to energy levels rather than the excitation mechanisms.

Page 1, right column bottom paragraph: Insert(ICCD) camera is often used to.....

The ICCD is not the only type of detector used in LIBS, today I would say that standard low-cost CCD spectrometers are more common.

Page 2, section 2: I think that the introductory text should include a short explanation of what a frequency comb is. You cannot simply assume that all interested readers know that this is in fact a laser system with a special output in the frequency domain. There are references included, but for very special terminology of this kind I think a brief description in the text is warranted. Likewise, an explanation that fs stands for femtosecond would be helpful.

Page 3, top left column: What is meant by beams gently focused? Please rephrase this to something more stringent!

The word "uncalibrated" to describe the mica sample is also bad terminology. A reference material (RM) is not described as "calibrated", what is meant here is that it has not been analysed for chemical content by other methods. Rephrase! It is also an odd statement that the sample is "believed to contain < 3% of Rb". Maybe approximately 3%? At least try to be a bit more specific!

Finally, I think it is better English to write "We recorded the full interferogram...", since the rest of the experimental description is written in the past tense.

Reviewer #3 (Remarks to the Author):

This is the first report to demonstrate that it is possible to measure D_{1} and D_{2} absorption lines of potassium and D_{2} line of rubidium simultaneously by adopting dual-comb spectroscopy for absorption spectrum measurements of laser induced plasma (LIP). Dual-comb spectroscopy itself has already been utilized in molecular spectroscopy; this study has introduced this new spectroscopic technique to the measurement of LIP for the first time. Currently, the techniques applied to the LIP measurement include LIBS for measuring the emissions from thermally excited monatomic species, and LAS and LIF for resonant excitation of the species in ground and metastable states. However, the method used in this study combines the advantages of both spectroscopic techniques and thus it is thought to be innovative. I think that this technique could have a large impact on future LIP study, and thus I recommend publication of this study in Nature Communications.

However, as there are some errors and minor points requiring reconsideration in the current version of the manuscript, revision is desirable before publication.

[1] (Abstract, Chapter 1) Although the authors state that the wavelength range of this method is similarly wide in LIBS, using the laser described here the spectrum is measured in the range of 764 to 780 nm (≈ 16 nm), which is much narrower than that of LIBS. Is this wavelength band sufficient for the assumed application? By changing the laser, how wide a wavelength range can be covered?

[2] (Figure 3-4, Chapter 3) What are the delay time and time window when acquiring the interferogram data of the spectra in figure 3? Are they the same as figure 2 (220 μ s, 440 μ s)?

Is it possible to reduce the noise in Fig. 3 (b) as small as that in Fig. 4 by increasing the number of averaged ablation shots?

[3](Chapter 3, optional) It seems that a gate-time width of about microseconds is necessary for the time-resolved measurement of the LIP which the authors plan as a future work, but is there any prospect of obtaining a sufficient S/N ratio?

[4] (Figure 1 caption) There are no dashed lines indicating feedback loop.

[5] (Figure 1) It is necessary to add the explanation of symbols in the figure, i.e. "S", "G" in caption.

[6] (Figure 2) The wavelength number of upper right of this figure "764.1" should be changed by "764.0".

[7] (Fig.2-4) It is preferable to add information of each peak, like $^{85}\text{Rb } F=1/2$ etc.

[8] (Figure 2) Why does this magnitude of back ground noise depend on wavelength?

[9] There are some typographical errors.

(p.1, left column) sufficiently

(p.1, left column) planarary

(p.1, right column) utilize

(p.2, right column) semic-static

(p.3, right column) demonstates

(p.4, left column) severely

(p.4, left column) occurs

(p.4, right column) generation

(p.4, right column) demonstrated

Manuscript Revisions

Reviewer #1

- *1. A short proceedings publications by the same authors, Jenna Bergevin et al, "Dual-Comb Spectroscopy of Laser-Induced Plasmas" in CLEO 2017 (SW1L.2), contains very similar context as the current paper, including the same title. I am concerned about the noveltys of the work, and if copyright issues may arise.*

The prior submission to CLEO describes related but preliminary work. The CLEO conference publishes extended abstracts and it is not considered as a journal article/communication. We believe there is no issue about copyright violation or previous publication.

- *2. This is a communication and the paper will not be long, but the author spend very little time addressing the underlying physics and chemistry and instead focuses the previous literature and technology potential. Importantly the data interpretation seems rushed. This paper would benefit from a more extended discussions of the Rb D2 absorption spectra.*

The primary role of the Rb transitions is to demonstrate that we can in fact resolve the narrow ground state hyperfine and isotopic shifts at a level comparable to that of a room temperature gas cell, while still measuring the K lines. On the contrary, because these transitions are so well characterized already it makes an excellent test system to demonstrate this new technique. We have added labels to the absorption peaks in figures 2-4 to indicate the specific transitions in Rb which we are measuring. In future works, we plan to apply the DCS spectroscopy tool reported in the present manuscript to study specific aspects of plasma physics and chemistry in greater detail.

Reviewer #2

- *Page 1, middle left column: In my opinion, the word "thermally-excited" is incorrect terminology. ...I therefore suggest using "electronically excited" throughout the text.*

Manuscript changed as suggested.

- *Page 1, right column bottom paragraph: Insert ...(ICCD) camera is often used to... The ICCD is not the only type of detector used in LIBS, today I would say that standard low-cost CCD spectrometers are more common.*

Manuscript changed as suggested.

- *Page 2, section 2: I think that the introductory text should include a short explanation of what a frequency comb is....*

We agree and thank the reviewer for the reminder to broaden the discussion for a wider audience. We have added a short paragraph explaining the concept of the frequency comb as it relates to the current manuscript. See revised text for details.

- *Page 3, top left column: What is meant by beams gently focused? Please rephrase this to something more stringent!*

We have modified this sentence to be more specific as suggested. The sentence now reads:

"...one pair of overlapped beams focused into a vacuum chamber using a 20 cm focal length lens just above the surface of either a NIST glass sample..."

- *The word "uncalibrated" to describe the mica sample is also bad terminology. A reference material (RM) is not described as "calibrated", what is meant here is that it has not been analysed for chemical content by other methods. Rephrase! It is also an odd statement that the sample is "believed to contain < 3% of Rb". Maybe approximately 3%? At least try to be a bit more specific!*

We have modified this sentence to be more specific as suggested. The sentence now reads:

"the mica sample had not been analyzed for chemical content, but contained approximately 3% or less of Rb."

- *Finally, I think it is better English to write "We recorded the full interferogram...", since the rest of the experimental description is written in the past tense.*

Manuscript changed as suggested.

Reviewer #3

- [1] (Abstract, Chapter 1) Although the authors state that the wavelength range of this method is similarly wide in LIBS, using the laser described here the spectrum is measured in the range of 764 to 780 nm (≈ 16 nm), which is much narrower than that of LIBS. Is this wavelength band sufficient for the assumed application? By changing the laser, how wide a wavelength range can be covered?

This is a good comment as we are specifically comparing dual-comb absorption spectroscopy to current LIBS techniques. The Ti:sapphire lasers in the current experiment utilized spectral bandwidths of around 15-20nm each (see Methods section). These lasers are known to easily produce bandwidths exceeding 100 nm. However, in this initial work our goal was to investigate the Rb D2 line as well as both K lines so we did not need to utilize dispersion compensation techniques to further broaden the fs laser spectrum. In future work we can demonstrate a broader spectral coverage. The use of much broader spectral coverage has been demonstrated in previous dual-comb using static gas cells. Therefore, increasing spectral coverage while still providing similar spectral resolution and s/n will be straightforward.

We had already partially addressed this in the conclusions paragraph, where we state:

“The spectral coverage of DCS can be easily extended to almost any portion of the electromagnetic spectrum. For example, with use of nonlinear optical fibers for supercontinuum generation, DCS laser systems can measure optical spectra spanning 100's of nm \cite{Okubo2015}. Further spectral coverage can be achieved with sum \cite{Potvin2013} and difference frequency generation\cite{Schliesser2012}, and even intra-cavity high harmonic generation\cite{Jones2005, Gohle2005} to the vacuum and extreme-ultraviolet.”

To make this point clearer, we have added additional text in the introduction. On page 2, beginning in section 2 “Dual-comb spectroscopy”, we have modified the first paragraph to give more background on the fs frequency comb (as requested by Reviewer #2). In this paragraph we now state:

“... Given the short pulse durations that can be obtained directly from many laser systems (<10 fs), the bandwidth of these optical frequency combs can easily extend beyond 100 nm. Furthermore, thanks to the high peak powers provided by the pulses, nonlinear frequency conversion can be utilized to extend the frequency comb coverage to over an optical octave. This makes the achievable spectral coverage comparable to that found in LIBS.”

- [2] (Figure 3-4, Chapter 3) What are the delay time and time window when acquiring the interferogram data of the spectra in figure 3? Are they the same as figure 2 (220 μ s, 440 μ s)?

The delay time was the same. However, the time window was longer for that data set, 1.5ms, which corresponds to the 0.18GHz resolution noted in Fig. 3. This is similar to that used to record the spectrum from the Rb cell alone, as shown in Fig. 4, where additional averaging was done. This time window was substantially longer than the plasma lifetime, and likely not optimum, but it was the time window used for that data set. Because of this, the delay time may not have as much significance, and we did not directly cite it. As mentioned in the manuscript, reduction in s/n and spectral resolution due to the long data acquisition times are somewhat reduced as the plasma evolution is much less dramatic after the first 100 microseconds.

- Is it possible to reduce the noise in Fig. 3 (b) as small as that in Fig. 4 by increasing the number of averaged ablation shots?

Yes, it is possible to reduce the noise seen in Fig. 3(b) simply by averaging more as we did in Fig. 4. It is also possible to do this for the laser ablation data (Fig 3a) to comparable levels by averaging multiple ablation shots. However, this would require a much longer data acquisition time, given that the data for Fig. 3 was taken at the 10Hz repetition rate of the ablation laser. The accuracy of the line shape would depend on the ablation conditions being repeatable, and the laser would need to be stabilized to an absolute frequency in order to prevent drift over the longer data acquisition time. These are all achievable with improvements to the current system. However, the focus of this initial work was on showing the ability to obtain the needed spectroscopic information in a single, or few, laser ablation shots.

- [3](Chapter 3, optional) It seems that a gate-time width of about microseconds is necessary for the time-resolved measurement of the LIP which the authors plan as a future work, but is there any prospect of obtaining a sufficient S/N ratio?

It is true that with the current configuration the s/n and resolution would suffer going directly to the 1-10 microsecond range. As we mention in the text, the s/n in the current experiment could be greatly improved using simple techniques already demonstrated in other DCS experiments (e.g. back-ground reference beam interleaved between ablation shots). A direct route to improved temporal resolution would be to utilize a laser system with a higher repetition rate. A sentence has been added to the text to address this. There are more complex approaches that may also improve the temporal resolution without reducing the s/n, but these are untested new concepts, which we intend to address in future publications.

We modified the last sentence on the last page, just before the conclusions, to read:

Probing the plasma with increased time-resolution utilizing higher repetition rate laser sources can be helpful in future work to study the dynamic evolution of the ionic, atomic, and molecular species within the plasma.

- [4] (Figure 1 caption) There are no dashed lines indicating feedback loop.
Figure changed as suggested.
- [5] (Figure 1) It is necessary to add the explanation of symbols in the figure, i.e. "S", "G" in caption.
Figure changed as suggested.
- [6] (Figure 2) The wavelength number of upper right of this figure "764.1" should be changed by "764.0".

The axis is actually correct given the number of significant digits shown. Since the lower axis is plotted as a linear function of frequency, the upper axis is not exactly linear since it is a function of wavelength.

- [7] (Fig.2-4) It is preferable to add information of each peak, like ^{85}Rb F=1/2 etc.
Figure changed as suggested.
- [8] (Figure 2) Why does this magnitude of back ground noise depend on wavelength?

This is a result of the finite bandwidth of the laser spectrum. With lower power levels in the wings of the spectrum, the s/n ratio decreases. Since we are plotting absorbance, this results in an increase of the background noise levels in the wings of the optical spectrum.

- [9] There are some typographical errors.

The following typographical errors have all been fixed, and the manuscript double-checked for any additional typos.

- x (p.1, left column) sufficiently
- x (p.1, left column) planarary
- x (p.1, right column) utilize
- x (p.2, right column) semic-static
- x (p.3, right column) demonstates
- x (p.4, left column) severly
- x (p.4, left column) occurs
- x (p.4, right column) genertion
- x (p.4, right column) demonstated

REVIEWERS' COMMENTS:

Reviewer #3 (Remarks to the Author):

I confirmed that all of my comments on the previous manuscript have been addressed in the new version manuscript. From the author's reply to my question, I am convinced that the author's method is a new spectroscopic technique with potential possibilities to replace LIBS. I am willing to recommend publishing this paper in Nature Communication.

We are pleased with the response we have received from the reviewers as shown below.

REVIEWERS' COMMENTS:

Reviewer #3 (Remarks to the Author):

I confirmed that all of my comments on the previous manuscript have been addressed in the new version manuscript. From the author's reply to my question, I am convinced that the author's method is a new spectroscopic technique with potential possibilities to replace LIBS. I am willing to recommend publishing this paper in Nature Communication.